# Experimental and Clinical Investigation of Cytokines in Migraine: A Narrative Review

**DOI:** 10.3390/ijms24098343

**Published:** 2023-05-06

**Authors:** Gaku Yamanaka, Kanako Hayashi, Natsumi Morishita, Mika Takeshita, Chiako Ishii, Shinji Suzuki, Rie Ishimine, Akiko Kasuga, Haruka Nakazawa, Tomoko Takamatsu, Yusuke Watanabe, Shinichiro Morichi, Yu Ishida, Takashi Yamazaki, Soken Go

**Affiliations:** Department of Pediatrics and Adolescent Medicine, Tokyo Medical University, Tokyo 160-0023, Japan; kanako.hayashi.0110@gmail.com (K.H.); sunflowernk69@gmail.com (N.M.); jerryfish_mika@yahoo.co.jp (M.T.); chiako0930@gmail.com (C.I.); shin.szk@gmail.com (S.S.); rie.ishimine@med.tohou.ac.jp (R.I.); akikokasuga@outlook.com (A.K.); haruka.komkom@gmail.com (H.N.); t-mori@tokyo-med.ac.jp (T.T.); vandersar_0301@yahoo.co.jp (Y.W.); s.morichi@gmail.com (S.M.); ishiyu@tokyo-med.ac.jp (Y.I.); tyamaz@tokyo-med.ac.jp (T.Y.); soupei59@gmail.com (S.G.)

**Keywords:** migraine, neuroinflammation, cytokines, IL-1β, TNF-α, mouse models, cortical spreading depolarization

## Abstract

The role of neuroinflammation in the pathophysiology of migraines is increasingly being recognized, and cytokines, which are important endogenous substances involved in immune and inflammatory responses, have also received attention. This review examines the current literature on neuroinflammation in the pathogenesis of migraine. Elevated TNF-α, IL-1β, and IL-6 levels have been identified in non-invasive mouse models with cortical spreading depolarization (CSD). Various mouse models to induce migraine attack-like symptoms also demonstrated elevated inflammatory cytokines and findings suggesting differences between episodic and chronic migraines and between males and females. While studies on human blood during migraine attacks have reported no change in TNF-α levels and often inconsistent results for IL-1β and IL-6 levels, serial analysis of cytokines in jugular venous blood during migraine attacks revealed consistently increased IL-1β, IL-6, and TNF-α. In a study on the interictal period, researchers reported higher levels of TNF-α and IL-6 compared to controls and no change regarding IL-1β levels. Saliva-based tests suggest that IL-1β might be useful in discriminating against migraine. Patients with migraine may benefit from a cytokine perspective on the pathogenesis of migraine, as there have been several encouraging reports suggesting new therapeutic avenues.

## 1. Introduction

The role of neuroinflammation in the pathophysiology of migraine is being increasingly recognized [1,2]. Neurogenic inflammation in migraine is known to cause the release of neuropeptides such as calcitonin gene-related peptide (CGRP) and substance P, mainly from the trigeminal nerve, leading to arterial vasodilation, plasma protein extravasation, and mast cell degranulation [3,4]. These series of responses possibly involve inflammatory cytokines and chemokines [1,2,5]. Cortical spreading depolarization (CSD), a recognized mechanism underlying migraine with aura, causes meningeal inflammation by activating macrophages and mast cells and promoting the production of multiple inflammatory cytokines in experimental animals [6,7,8,9,10].

From a clinical perspective, non-steroidal anti-inflammatory drugs can effectively prevent migraine attacks [11]. In contrast, steroids with anti-inflammatory properties have been reported to negatively affect amelioration [12] and short-term prevention [13,14] of acute migraine attacks. The exact mechanism underlying neurogenic inflammation in migraines remains unknown.

This review surveys the current literature on neuroinflammation in the pathogenesis of migraine, particularly from a cytokine perspective, and identifies new research directions on therapeutic interventions for migraines targeting neuroinflammation.

## 2. Summary of Search

Focusing on cytokine- and chemokine-mediated inflammatory reactions, we present a narrative review on neuroinflammation and pathogenesis of migraine headaches. A literature search was conducted using the PubMed database, which included articles published until December 2022. The keywords used in the search were “migraine” and “inflammation” or “neuroinflammation” or “cytokine” or “chemokine.”

## 3. Experimental Study on Neuroinflammation

Several models have been used to investigate the relationship between migraine and neuroinflammation, primarily mouse models with CSD and nitroglycerin (NTG)-induced rat models but also transgenic mouse models [15], inflammatory soup models [16,17], and the sound stress models [18].

### 3.1. CSD Mouse Model of Migraine

CSD is an underlying mechanism of migraine with aura and is a wave of neuronal depolarization with glial and vascular activation [19,20]. CSD mice have been used as experimental models for the pathophysiology of migraine. CSD may cause a substantial inflammatory response by opening pannexin-1 megachannels in neurons, activating caspase-1, releasing high-mobility group box 1 from neurons, and activating astrocytes [21]. Following CSD, the inflammasome complex is rapidly formed, as evidenced by the appearance of a truncated form of caspase-1 in neurons and IL-1β release in the cerebrospinal fluid (CSF) [21]. Previously, CSD has also been shown to induce proinflammatory cytokines such as IL-1β, IL-6, and TNF-α in a mouse migraine model in vitro [10,22] and in vivo [8,23]. However, all of these studies utilized highly invasive and potentially harmful methods to induce CSD, such as craniotomy. Brain injury and CSD activate an acute inflammatory response in the brain parenchyma [24,25]. Therefore, the intensity and time course of the inflammatory response measured after CSD may have contributed to the inconsistent data, owing to the invasive CSD induction method [26]. A recent comparative study of brain tissue from the invasive and non-invasive sides induced by CSD revealed a markedly different inflammatory response [27]. To address this concern, studies comparing the expression levels of target genes in injured and intact cortical tissue after a single CSD episode [27] and non-invasive optogenetic CSD [28] have emerged and yielded consistent data for inflammatory modifications [27,28]. CSD induced over 1 h significantly increased IL-1β expression [27,28]. Subsequently, TNF-α was expressed at 1–4 h, while IL-6 expression was not observed [27,28]. Several studies have also reported the increased mRNA expression of TNF-α and IL-1β [8,26,29]. Particularly, the intense upregulation of TNF-α compared to IL-1β 4 h after CSD has also been confirmed in other experimental studies [8]. In a recent study comparing injured and intact cortical tissue after a single CSD, the expression of IL-1β, TNF-α, and IL-6 was stronger at the injury site than in the intact cortex affected by CSD. Interestingly, while TNF-α and IL-1β but not IL-6 were upregulated in the intact cortex, the most affected cytokine was TNF-α. Furthermore, CGRP and Panx1 were strongly upregulated in the intact cortex affected by CSD [27]. This study suggests different roles for CSD in the development of neuroinflammation in injured and intact neuronal tissues, and that TNF is the inflammatory cytokine most affected by CSD and is closely related to CSD [27].

Regardless of IL-4 and IL-10, in a study using microsphere-based flow cytometric immunoassays and hippocampal organotypic cultures (HOTCs), CSD caused significant increases in IL-4 and IL-10 in HOTC after 6 h [22]. IL-10 was found to be significantly higher in HOTC at 6 h after CSD but significantly lower in the surrounding medium at that time and 1 day later [22]. The marked decrease in IL-10 in the medium at 6 h and 1 d after CSD was unlikely to be caused by decreased production, as the absolute levels in all groups were above baseline. A significant increase in IL-4 production in rat spleen has also been observed in CSD [9]. Previous in vitro [30] and in vivo [23] reports have indicated no evident increase in IL-4 and IL-10, and recent noninvasively induced [28] and invasive unilateral CSDs with [27] did not address these anti-inflammatory cytokines.

Undoubtedly, CSD affects inflammatory cytokines and CGRP, but CSD has also been suggested to be associated with other central diseases apart from migraine. However, the extent to which CSD and these neuroinflammatory mechanisms are deeply involved in the pathogenesis of migraine remains unknown. (Further, suppressing inflammation by CSD does not necessarily lead to the treatment of migraines.)

### 3.2. Spontaneous Migraine-Like Mouse Model by Nitroglycerin

NTG is a well-established and reliable experimental clinical and preclinical approach for studying migraine headaches. Although NTG can induce a migraine-like phenotype in rodent models, it is unclear whether NTG and its degradation product, nitric oxide, are determinants of the underlying pathophysiology of migraine [31]. On the other hand, previous reports in a rat model of meningeal inflammation after NTG have demonstrated increased inflammatory cytokines, IL-1β in the dura mater and IL-6 in dural macrophages and CSF [32,33]. Recently, there have been studies of acute migraine models with a single injection of NTG and chronic migraine models with continuous NTG administration, with the latter being more common [34,35,36,37], and some studies have compared both models [38,39]. Both models demonstrated increased gene expression of proinflammatory cytokines IL-1β, IL-6, and TNF-α and reduced anti-inflammatory IL-10 in both peripheral (i.e., trigeminal ganglia, TGs) and central sites (i.e., trigeminal nucleus caudalis, TNC) [34,35,36,37,38,39]. These inflammatory changes were not evident in the acute and chronic phases, but differences in glial cell activation were observed [39]. Most chronic model studies have documented increased microglial activation in the TNC region [35,36,37,40], while others have found increased TG satellite glial cell reactivity without change in microglial activation [39]. The lack of microglial activity in the chronic phase may be due to several factors: (i) markers used for microglial labeling, (ii) species, (iii) dose of NTG, and (iv) time of ex vivo evaluation since the last NTG injection [39]. Glial cells repeatedly but intermittently exposed to NTG may respond by changing their phenotype more rapidly and dynamically [41]. As the authors also noted, depending on the timing of microglial observation, their activation might not have been captured [39].

Elevated satellite glial cell reactivity in the chronic phase, which is not observed in the acute phase [39], and fluctuations in inflammatory cytokines present in the acute and chronic phases may indicate a persistent inflammatory state in the pathogenesis of migraine. Longer procedures and stronger triggers may be required to activate satellite glial cells and express proteins [42]. In other words, prolonged and intense stimuli can activate satellite glial cells, making migraine chronic, and activating satellite glial cells might indicate chronic migraine.

### 3.3. Inflammatory Soup Model of Migraine

An inflammatory soup containing prostaglandin E2, serotonin, histamine, and bradykinin in the dura mater administered to the meninges causes enlargement of the mechanoreceptive field of the skin, sensitization of nociceptors to mechanical stimuli, and sensitization of caudal neurons of the trigeminal nucleus in anesthetized animals [43]. Inflammatory soup models have successfully elucidated the mechanistic relationship between immune mediator-induced activation of meningeal nociceptors and associated changes in trigeminal sensory processing, and several studies have used inflammatory soup in mouse models of migraine [16,34,44,45]. This mouse model has also been shown to upregulate the expression of cytokines IL-1β and IL-18 [16,34,45]. A study showed that microglia–astrocyte crosstalk via IL-18/IL-18R might lead to microgliosis and astrogliosis in the dorsal horn of the medulla following repeated dural inflammatory soup (IS) stimulation, resulting in migraine-like behavior [16]. IL-18 dural infusion induced nociceptive behavior and glial activation; IL-18 was the product of activation of microglial toll-like receptor 4, acting on IL-18R expressed in astrocytes and subsequently activated nuclear factor-κB (NF-κB), which in turn led to the activation of astrocytes [16]. Furthermore, repeated dural IS stimulation triggered upregulation of the P2X7 receptor (P2X7R) and the activation of NLRP3 inflammasome with proinflammatory cytokines (IL-1β and IL-18) release. Furthermore, active caspase-1 also cleaved full-length Gasdermin D (GSDMD-FL) to N-terminal Gasdermin D (GSDMD-NT), leading to the pyroptotic cell death pathway.

Gliosis (microgliosis and astrogliosis), neuronal loss, and cognitive impairment also occurred. These pathological changes in the cerebral cortex might link to migraine-related cognitive impairment [45]. P2X7R, a member of the purinergic receptor family, has recently been implicated in the pathogenesis and progression of migraine [46], and P2X7R is elevated in animal models and patients with migraine [47,48]. With P2X7R upregulation in the TNC in other NTG-induced migraine mouse models [35], P2X7R has been found to promote CSD transmission [49,50] and identified as a potential therapeutic target for migraines. The inhibition of these cascades by P2X7R antagonists [51] and the prevention of inflammasome complex formation [34] could provide a novel migraine treatment strategy (Figure 1).

### 3.4. Sound Stress Migraine Model

Stress is the most widely known trigger of migraines. Although there are mouse models of stress due to repetitive restraint stress paradigms, social defeat, chronic variables, and acute or chronic stress from childhood stress [52,53,54], sound stress has been noted as a possible relevant mediator of headache induction [55], and mice with repetitive sound stimulation have emerged [18]. Unlike previously described models, this sound stress model successfully produced migraine-like behavior through unpredictable sound-mediated stress without dural stimulation or drug administration. This mouse model has also exhibited elevated plasma IL-6, TNF-α, and CGRP levels but not IFN-γ, IL-2, IL-4, IL-10, and IL-17 levels [18]. Further, females expressed higher levels of TNF-α, IL-6, and CGRP than males. CGRP has been implicated in the release of proinflammatory cytokines [18], which sensitize nociceptors in the dural meninges [56]. IL-6 application in the dura and cisterna has been shown to cause periorbital and hindlimb skin hypersensitivity, triggering migraine-like symptoms [57].

Although migraine headaches are three times more common in women during their reproductive years than in men, the mechanism of sex differences in migraine is still, in fact, largely unexplored [58]. Although sex differences mediated by sex hormones have been investigated in mouse migraine models [58,59], these studies have not evaluated inflammatory reactions. Pain mechanisms may differ between men and women due to inflammatory responses [60]. Therefore, these differences in inflammatory responses might contribute to the sex differences in migraine headaches. However, it is unclear whether these inflammatory changes are linked to migraine headaches or whether the differences in inflammatory markers are due to migraine headaches. Table 1 summarizes the results of the study.

## 4. Clinical Research on Neuroinflammation

Many reports are accumulating on the inflammatory response, particularly involving cytokines, in patients with migraines. Compared to the aforementioned mouse models, studies continuously examining inflammatory markers are limited; however, studies have been conducted during seizures and the interictal period. The main inflammatory cytokines investigated were TNF-α, IL-6, and IL-1β, while the anti-inflammatory cytokines were IL-10 and IL-4. This was summarized in a systematic review by Thuraiyah et al. [62]. The results are summarized in Table 2.

### 4.1. Examination of Serum or Plasma Cytokines during Migraine Attacks in Patients with Migraine

Most serum or plasma studies during migraine attacks have primarily reported unchanged TNF-α levels, with inconsistent results for IL-1β and IL-6 levels. In examining TNF-α, many studies during migraine attacks have reported unchanged levels compared to the interictal period [63,64,65,66,67,68], while others confirmed significant differences [69,70]. Many reports have not confirmed a significant increase in IL-6 levels [64,67,69,71,72] and only two confirmed significant differences [65,70]. In the study of IL-1β, the number of reports with confirmed significant differences [69,70,73] and those with no significant increase [63,65,67,74] were similar.

The number of studies on anti-inflammatory cytokines, mainly IL-4 and IL-10, are fewer and more limited than the number of studies on inflammatory cytokines. In studies of non-inflammatory cytokines in patients with migraines, some reports indicated lower [64,75,76] or unchanged [69] IL-4 levels compared with during attacks. In studies of IL-10 in migraineurs during non-seizures, there are reports of increased IL-10 compared to during attacks [65,69] and unchanged IL-10 compared to [67].

Factors that modify cytokines’ levels include the number of samples, measurement methods, and the broad clinical spectrum. However, it is also difficult to clearly distinguish between migraine attacks and non-attacks [77,78,79]. For example, in a report by Yücel et al. confirming significant findings of the inflammatory cytokines TNF-α and IL-6, sampling timing was strictly set to within 3 h from the onset of a migraine attack before treatment. Significant findings were obtained despite the relatively small sample size (*n* = 24), which may indicate the importance of timing.

Considering this information, continuous analysis of cytokines is valuable and yields highly esthetic and consistent data.

Sarchielli et al. performed two serial analyses of cytokines in the jugular venous blood [64,80]. In 8 patients with migraine without aura, catheters were inserted within 30 min of a migraine attack, and internal jugular venous blood samples were obtained immediately after insertion, 1, 2, and 4 h after attack onset and 2 h after cessation of attack to examine cytokinesis. These studies demonstrated that significant increases in IL-6 and TNF-α were observed in parallel within 2 h of seizure onset, whereas IL-1β increased slightly from 1 to 4 h but then decreased, reaching values at seizure termination [64]. The IL-4 levels decreased from 1 to 4 h after catheter insertion. At the end of the seizure, the IL-4 levels returned to their value at the time of seizure onset [64]. CGRP significantly increased after 1 h, and IL-8 reached its highest level after 4 h. The levels of the other two chemokines, RANTES (CCL5) and MCP-1, did not significantly change at any time [80]. Unlike the data from jugular venous blood, the levels of cytokines in the peripheral blood of patients with migraine did not change at any time point in the study [64,80].

Most blood measurements from the internal jugular vein reflect circulation in the brain parenchyma [81]. From an ethical standpoint, obtaining samples from the internal jugular vein is an invasive and infeasible procedure. These results are extremely valuable for understanding the cytokine dynamics in patients with migraine.

### 4.2. Examination of Serum or Plasma Cytokines during the Interictal Period in Patients with Migraine

Most studies investigating the interictal phase of TNF-α have reported elevated levels compared to controls [70,82,83]. However, some interictal studies have also shown unchanged levels compared to controls [67,68,72,79,84,85,86,87]. Similarly, many studies have reported elevated IL-6 levels compared to those in healthy controls [64,65,67,70,71,72,82,88,89,90,91], while some have shown unchanged levels compared to controls [69,72,79,84,85,92]. Most interictal investigations of IL-1β show no change compared to controls [69,72,82,83,84,87], while some reports have shown increased levels [67,70,93].

Regarding the comparison between episodic and chronic migraines, two studies examined differences in cytokine levels between patients with episodic and chronic migraines [90,94]. Although TNF-α levels were similar between patients with episodic and chronic migraine, both episodic and chronic migraine patients had significantly higher median TNF-α levels compared to control subjects [94]. Furthermore, a positive correlation was observed between TNF-α levels and the probability of having migraine headaches in multivariate regression models, even after accounting for sex, age, body mass index, and dietary intake of energy, carbohydrates, protein, lipids, and mono- and polyunsaturated fatty acids [94]. The other study presented that serum IL-6, CRP, and TNF-α were significantly higher in subjects who developed chronic migraines compared to episodic migraines and controls, and a positive correlation between headache frequency and serum IL-6 and TNF-α was confirmed [90]. Although studies examining differences in cytokine levels between episodic and chronic migraine patients have not led to a definite consensus, it is interesting to note that biomarkers, including inflammatory cytokines, are associated with headache frequency and severity [90,95], even though headache is a subjective symptom.

Most studies have reported decreased levels of IL-10 in the interictal period in migraine patients compared to controls [67,79,83,89,93]. Some consider the level invariant [65,69], with only one study reporting an increase in levels of IL-10 [87]. Interictal studies on IL-4 levels in migraine have yielded few reports with no certainty [65,69,93,96].

### 4.3. Cerebrospinal Fluid (CSF) Analysis of Patients with Migraine

Investigations on inflammatory biomarkers in the CSF of migraine patients are limited compared to those in the serum and plasma, as described above, and three studies investigated cytokine levels in the CSF [77,78,79] and presented some interesting findings. During migraine attacks (ictal period), elevated TNF-α levels in CSF have been observed in patients with chronic migraine despite normal serum TNF-α levels [77]. Another study demonstrated that IL-1β antagonist, MCP-1, and TGF-b1 were significantly increased in the CSF of patients with episodic tension-type headache (TTH) and migraine with and without aura compared to those without pain [78]. However, no significant differences in cytokine levels were found between the headache types. Interestingly, a result not seen in migraine is that the inflammatory cytokine MCP-1 correlates with the anti-inflammatory cytokines IL-1ra, TGF-b1, and IL-10 in episodic TTH [78]. Determining whether these cytokine changes are migraine-specific, whether they represent an inflammatory response to headaches, and whether these changes affect the blood–brain barrier (BBB) remains unknown.

Recently, Cowan et al. published a detailed report where blood and lumbar CSF samples from 42 migraine patients, including CM and EM, were evaluated for two conditions: headache attack before relief therapy and migraine-free for at least 48 h (interictal) compared to 14 healthy volunteers [79]. They found no significant difference in the levels of inflammatory markers IL-6, IL-8, IL-10, TNF-α, IFNγ, and CRP in CSF and plasma, though IL-10 levels were significantly lower in migraine. The timing of sample collection, which is usually a problem when examining markers such as cytokines, was set quite strictly, such as 2–8 h during headache attacks before treatment. The data appears highly trustworthy, although the number of cases may be insufficient. The study also demonstrated that the mean CSF but not plasma soluble vascular cell adhesion molecule-1 (sVCAM-1) levels in EM were significantly higher in those with more frequent headaches per month. Levels of sVCAM-1 in the CSF have been suggested as potential biomarkers for the high frequency of migraine headaches [79]. VCAM, one of the markers of endothelial function in BBB, is elevated; however, typical signs of endothelial barrier disruption have not been identified in this study [79]. Other typical markers suggestive of BBB disorders, such as MMP-9 [97], ICAM-1 [98,99], and VCAM-1 [99], were also elevated in blood obtained from migraine patients. In contrast, there are several negative findings regarding BBB disorders from recent MRI studies [100,101], and no clear conclusions regarding migraine and BBB disorders in humans have been reached.

The correlation between peripheral and central cytokine levels and whether CSF measurements provide a better cytokine profile of migraine patients compared to peripheral measurements cannot be evaluated. Considering invasiveness, there is little likelihood that spinal fluid will be considered in the future.

### 4.4. Saliva Study of Patients with Migraine

In contrast to spinal fluid studies, two recent studies used saliva, which can be obtained noninvasively [92,102]. The interictal investigation of saliva also showed unchanged levels of IL-1β [92,102]. Interestingly, the saliva levels of IL-1β increased after cervical non-invasive vagus nerve stimulation therapy (nVNS), yielding 2.5 times higher values than those measured in healthy controls [92]. The authors suggested that these facts might suggest an ongoing inflammatory process, as some subjects who underwent nVNS still had headaches despite clinical improvement. Another research reported findings of decreased IL-1β and elevated IL-6 levels during headache attacks [102] and also demonstrated that IL-1β had the highest discriminative value between patients with headaches and controls compared to CRP and IL-6 [102]. Reports of saliva testing for migraine are limited; therefore, extensive comparisons, such as with blood testing, could not be made. However, it is expected to develop in the future, partly because it is non-invasive.

Non-invasive saliva testing may allow for temporal sampling and evaluation of the association between CSD and inflammatory cytokines in humans, even during a limited period of migraine aura.

**Table 2 ijms-24-08343-t002:** Brief summary of clinical research on neuroinflammation.

Examination of serum or plasma cytokines during migraine attacks in patients with migraine	No changes in proinflammatory cytokine TNF-α levels have been reported in serum or plasma during migraine attacks [63,64,65,66,67,68]; results for IL-6 and IL-1β levels are inconsistent. Lower or unchanged anti-inflammatory cytokine, IL-4, levels, and increased or unchanged IL-10 levels during migraine attacks have been reported, with inconsistent results throughout. However, serial analyses of cytokines in jugular venous blood collected by catheterization during migraine attacks resulted in consistent data, an initial increase in IL-1β, followed by increases in IL-6 and TNF-α, and a decrease in IL-4, which returned to its initial value at the end of the attack [64]. This study revealed no change in cytokine levels in peripheral blood collected simultaneously [64].
Examination of serum or plasma cytokines during the interictal period in patients with migraine	Most studies examining the interictal period of TNF-α [70,82,83] and IL-6 [64,65,67,70,71,72,82,88,89,90,91] have reported higher levels compared to controls and no change with respect to IL-1β levels [69,72,82,83,84,87]. IL-10 levels during attacks in migraine patients have been reduced compared to controls [67,79,83,89,93], but IL-4 levels have been reported less frequently and without certainty.
Cerebrospinal fluid (CSF) analysis of patients with migraine	CSF studies are limited and not definitive, and a certain view of inflammatory and non-inflammatory cytokines has not been obtained. However, soluble vascular adhesion molecule-1 (sVCAM-1) in CSF, but not plasma, is higher with more frequent headache frequency, suggesting that sVCAM-1 levels in CSF might be a potential biomarker for frequent migraine and CM [79].
Saliva study of patients with migraine	There are markedly fewer saliva-based tests, and no certain view of cytokines has been obtained. IL-1β might be useful in discriminating against migraine [102].

## 5. Cytokines and Genetics

As mentioned above, animal and human studies suggest that TNF-α contributes to the pathogenesis of migraine, and that its secretory capacity might be regulated by a gene variant in the promoter region of the TNF gene. Guanine, located upstream of 308 base pairs in the TNF gene (NC_000006.12:g.31575254G>A), which is replaced by adenine, is thought to increase circulating levels of TNF-α [103]. Variants registered in dbSNP as rs1800629 have been noted as being associated with migraine, with some reports showing that migraine without aura is significantly more prevalent than in healthy populations [104,105,106]. There are also reports that found no significant differences [107,108]. A meta-analysis conducted by Schürks et al. in response to this discrepancy showed no overall association between rs1800629 and migraine. Subgroup analysis showed that the risk of migraine without aura due to this variant varied by ethnicity. The risk of migraine with aura was higher among Asians, and both effects were stronger in women [109]. Other meta-analyses showed that rs1800629 was associated with migraine overall in Asians. Subgroup analyses showed significantly higher odds ratios for variant retention in migraine with aura and no significant differences in migraine without aura [104]. It should be noted that the discrepancy in the results of these studies may indicate differences in the genetic background of migraine by race. An analysis of sites in linkage disequilibrium with variants including rs1800629 aimed at resolving previously reported discrepancies and showed that the LTA gene haplotype was associated with migraine risk in Koreans [110]. In addition, pharmacogenetics showed that the efficacy of NSAIDs was significantly lower in migraine attacks in rs1800629 carriers [111], suggesting that rs1800629 may be deeply involved in the pathophysiology of migraine attacks as well as in the risk of migraine. The reproducibility of the results is expected to be reported.

In a study of the IL-1α gene, holders of genotype T/T in the promoter region (889 base pairs upstream) have a significantly earlier onset of migraine than the wild types C/C and C/T, which are observed more frequently in migraine with aura [112]. There is a report of significant IL-1β gene variants [106], but no significant differences between migraine and controls for the promoter variants of the IL6 gene [113] and IL10 gene [114] have been determined. While the results from these genetic studies are very interesting, the details of the impact of the above variants on neuroinflammation in migraine remain unclear, as the genetic variants were not designed to allow examination of their impact on neuroinflammation.

With the widespread use of next-generation sequencers, many Genome-wide association studies (GWAS) have been conducted for complex genetic diseases. In migraine, GWAS have been performed to clarify its genetic background: in 2016, Gormley et al. explored disease susceptibility genes in 59,674 migraine patients and 316,078 controls in a meta-analysis of 22 GWAS studies [115]. They identified 38 genomic loci and showed that they were associated with genes expressed in blood vessels and smooth muscle. Among them, the TGFBR2 gene is interesting because it is a receptor for TGF-β and has implications in the neuroinflammatory pathway. The fact that this GWAS meta-analysis is limited to the genes most proximal to the relevant single nucleotide polymorphism (SNPs) opens the possibility that the distal genes are the true causative genes.

In 2022, Hautakangas et al. published a GWAS meta-analysis of 102,084 migraine cases and 771,257 controls that added data from four more GWAS studies to the Gormley et al. meta-analysis [116]. A total of 123 genomic loci were identified, 86 of which were novel locations. These included the CALCA and CALCB genes, which encode CGRPs targeted by migraine drugs, and the HTR1F gene, which encodes the 5-HT1F receptor. Interestingly, inflammation-related genes included the TGFBR2 gene as well as the MAPKAPK2 and PTK2 genes, which encode MK2, a downstream substrate of the p38MAPK signaling pathway that regulates apoptosis and inflammatory responses. It has been reported to promote the production of TNF-α and IL-6 [117]. Pyk2 encoded by the PTK2 gene is thought to activate MAPKs and NF-kB in response to both TNF-α and IL-1β [118]. Thus, it is suggested that Pyk2 may modulate neuroinflammation in migraine in association with TNF-α.

In these GWAS meta-analyses, all subjects were of European ethnicity. Therefore, caution should be exercised when applying the results of the meta-analyses to other racial groups. For example, the results may not be consistent with the observation that TNF gene variants are more common in Asian patients with migraine.

In human and mouse studies, there is no doubt that cytokine-mediated inflammatory phenomena occur during the pathogenesis of migraines.

## 6. Conclusions

In mouse models, CSD induces temporal alterations in inflammatory cytokines such as TNF-α, in particular, and in IL-1β and IL-6. However, it is also becoming evident that invasive procedures considerably influence the inflammatory response when CSD is induced. Various mouse models have been used to induce migraine attack-like symptoms. The spontaneous migraine-like mouse model using NTG examined episodic and chronic migraine and found increased satellite glial cell reactivity in the chronic phase that was not seen in the acute phase [39], suggesting that it may be an indicator of chronic migraine. In the inflammatory soup model of migraine, the release of IL-1β, IL-18, upregulation of P2X7R, and activation of the NLRP3 inflammasome have been implicated in IL-18/IL-18R-mediated microglial astrocyte crosstalk, and microglial astrocyte gliosis might be associated with migraine-related cognitive impairment [45]. In the sound stress migraine model, women expressed higher levels of TNF-α, IL-6, and CGRP than men [18], suggesting that differences in inflammatory response might be involved in the sex differences in migraine.

Compared with mouse studies, these data are less consistent in examining the inflammatory response to migraines in humans. Overall, serum or plasma studies of inflammatory cytokines during migraine attacks in patients with migraines have not provided a consistent picture. Research on serum or plasma during migraine attacks has reported no change in TNF-α levels and inconsistent results for IL-1β and IL-6 levels. However, in serial analyses of cytokines in jugular venous blood collected by catheterization during migraine attacks, consistent data have been obtained showing an initial increase in IL-1β, followed by increases in IL-6 and TNF-α, and a decrease in the anti-inflammatory cytokine IL-4, which returns to its initial value at the end of the attack. These results have not been obtained from peripheral blood collected simultaneously, indicating the complexity of cytokine studies in humans [64]. Most studies examining the interictal period of TNF-α and IL-6 have reported higher levels compared to controls and no change with respect to IL-1β levels. However, saliva-based tests have indicated that IL-1β might be useful in identifying migraine [102], and IL-1β might be a potent cytokine in the pathogenesis of the inflammatory response in migraine.

Genetic analysis of cytokines suggests that TNF-α and other cytokines are involved not only in the risk of migraine [104,105,106,119] but also in the pathophysiology during attacks [111]. Furthermore, comprehensive genetic analysis using GWAS has identified not only genes encoding CGRP and 5-HT1F receptors but also genes involved in the production of TNF-α and IL-6 and regulating apoptosis and inflammatory responses as candidate genes for migraine [115,116]. Although definitive results have yet to be obtained and prove a link to neuroinflammation via cytokines, progress in this field of genetic analysis is impressive, and future perspectives are promising.

Although the treatment approaches for migraine are not yet well established, IL-1 receptor antagonists suppress migraine attacks in patients with the cryopyrin-associated periodic syndrome, where mutations in the inflammasome component NLRP3 result in over secretion of IL-1β [120,121]. Furthermore, the usefulness of anakinra has already been reported in epilepsy [122,123,124], in which IL-1β is deeply involved in the pathogenesis [125], and which has been pointed out to have pathophysiological similarities with migraine [126,127]. Not only IL-1β but also TNF-α and IL-6, which are associated with migraine, as pointed out in this review, have already been established for clinical application in rheumatic diseases. The efficacy of each of these anti-cytokine therapies is also expected. As shown in Figure 1, mouse models suggest that P2X7R antagonists can inhibit these cascades [51] and prevent inflammasome complex formation [34], which will facilitate novel migraine treatment strategies. A cytokine perspective on migraine etiology may further aid the treatment of patients with migraine.

## Figures and Tables

**Figure 1 ijms-24-08343-f001:**
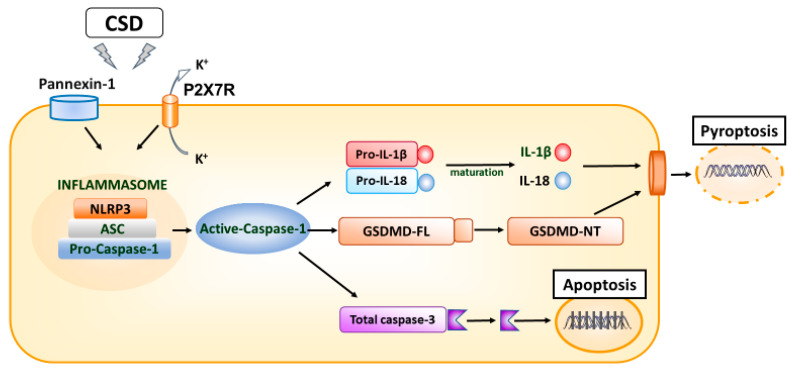
Schematic diagram of neuroinflammation in migraine. After the triggering factor causes CSD, the Pannexin-1 megachannel opens, P2X7R expression is elevated, and an inflammasome complex composed of NLRP3, an adaptor (i.e., ASC), and an effector protein (i.e., total caspase-1) is assembled. The formation of the inflammasome complex activates the total caspase-1, which is converted to active cleaved caspase-1. Activated caspase-1h can cleave both precursor IL-1β and IL-18 into the active inflammatory cytokines mature IL-1β and IL-18; it can also cleave and activate total caspase-3 to cleaved caspase-3 and induce apoptosis. Active-caspase-1 also cleaves GSDMD-FL to GSDMD-NT, provoking pyroptosis, a lysogenic type of cell death. In addition, it can cleave and activate total caspase-3 to cleaved caspase-3, leading to apoptosis. Suppression of these cascades with P2X7R antagonists or inhibition of inflammasome complex formation may lead to a novel migraine therapeutic strategy. Abbreviations: ASC—apoptosis-associated speck-like protein; GSDMD-FL—full length Gasdermin D; GSDMD-NT—N-terminal Gasdermin D; NLRP3—leucine-rich repeat pyrin containing protein-3.

**Table 1 ijms-24-08343-t001:** Summary of experimental study on neuroinflammation.

CSD migraine model	CSD induces temporal alterations in inflammatory cytokines such as TNF-α, IL-1β, and IL-6 [8,10,21,22,23,26,27,28,29]. On the other hand, it is also becoming evident that the invasion considerably influences the inflammatory response when inducing CSD. Some reports suggest an increase in the anti-inflammatory cytokines IL-4 and IL-10 [9,22], but this has not been investigated in recent non-invasive studies, and it is not possible to determine how anti-inflammatory cytokines are affected by CSD.
Spontaneous migraine-like mouse model using nitroglycerin	There are studies of an acute migraine model with a single injection of NTG [38,39] and a chronic migraine model with continuous NTG administration [34,36,37,61], and both models have shown increased gene expression of inflammatory cytokines IL-1β, IL-6, and TNF-α and decreased anti-inflammatory IL-10 [34,35,36,37,38,39]. An increase in satellite glial cell reactivity was detected in the chronic phase, suggesting that it may indicate chronic migraine [39].
Inflammatory soup model of migraine	In an inflammatory soup model of migraine, the expression of cytokines IL-1β and IL-18 increases with the upregulation of P2X7R [16,34,45]. Microglial and astrocyte gliosis associated with these inflammatory changes have been implicated in migraine-related cognitive impairment, and P2X7R has been suggested as a potential therapeutic target for migraine headaches [45].
Sound stress migraine model	The sound stress migraine model has been documented to have elevated plasma levels of IL-6, TNF-α, and CGRP but not IFN-γ, IL-2, IL-4, IL-10, or IL-17 levels [18]. Additionally, females expressed higher levels of TNF-α, IL-6, and CGRP than males [18], suggesting that differences in inflammatory response may be involved in the sex differences in migraine.

## Data Availability

The datasets generated and/or analyzed during the current study are available at the PubMed database repository (https://pubmed.ncbi.nlm.nih.gov/ (accessed on 1 April 2023).

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
