# Peer review of "Experimental and Clinical Investigation of Cytokines in Migraine: A Narrative Review"

_ijms, 2023, doi:10.3390/ijms24098343_

Round 1

Reviewer 1 Report

The authors in this review summarise the current knowledge regarding the role of cytokines in the pathogenesis of migraine. They have reviewed both various animal models and clinical results. More importantly, regarding the clinical studies, they have tried to reach a conclusion for the discrepancies found (for example when examining different body fluids). The authros have an extended experience in the field, and they have also cited both recent reseach articles, and important research which is not so contemporary but remains substantial for the field. 

My main problem here is that the authors do not discuss the role of lipids in migraine-related neuro-inflammation. I think that the perceived pain in migraine is mainly regulated by the release of different eicosanoids. In addition, i am sure that the authors are aware that the main molecules limiting acute inflammation from becoming chronic are the specialized pro-resolving mediators. I would suggest that the authors make an effort to connect the levels of cytokines with the metabolism of both pro-, and and anti-inflammatory lipids. Among these molecules there is a crosstalk, that includes metabolic (mainly in the plasma membrane) and trascriptional control. If authors do not find that these suggestions are under the scope of their review, then i would recomend that they change the title of their review and render clear that they do not discuss all the aspects of neuroinflammation, but solely the role of cytokines. 

Author Response

Manuscript ID: ijms-2357239

Type of manuscript: Review

Title: Experimental and clinical investigation of neuroinflammation in

migraine: A Narrative Review

To the Reviewer 1

We would like to express our sincere gratitude for the reviewers’ comments. We appreciate the time and effort that the reviewer has dedicated to providing insightful feedback. All revisions and additions are indicated as yellow-highlighted text in the manuscript. We sincerely hope that with these revisions, our manuscript will be suitable for publication in the International Journal of Molecular Sciences.

Reviewer: 1
Comments

My main problem here is that the authors do not discuss the role of lipids in migraine-related neuro-inflammation. I think that the perceived pain in migraine is mainly regulated by the release of different eicosanoids. In addition, i am sure that the authors are aware that the main molecules limiting acute inflammation from becoming chronic are the specialized pro-resolving mediators. I would suggest that the authors make an effort to connect the levels of cytokines with the metabolism of both pro-, and and anti-inflammatory lipids. Among these molecules there is a crosstalk, that includes metabolic (mainly in the plasma membrane) and transcriptional control. If authors do not find that these suggestions are under the scope of their review, then I would recommend that they change the title of their review and render clear that they do not discuss all the aspects of neuroinflammation, but solely the role of cytokines. 

Response:

Thank you for your valuable remarks. As you have pointed out, the role of lipids in the neuroinflammatory pathogenesis of migraine is critical. We recognize that migraine pain involves various eicosanoids and that cytokines are involved in the cross-talk, including metabolic and transcriptional regulation at the cell membrane. We have discussed the association of cytokine levels with pro-inflammatory and anti-inflammatory lipid metabolism. However, due to my own limited knowledge of lipid metabolism, I have revised the title of this review paper as follows:

Experimental and Clinical Investigation of Cytokines in Migraine: A Narrative Review

Reviewer 2 Report

In this manuscript titled “Experimental and clinical investigation of neuroinflammation in migraine: A Narrative Review,” the authors portrayed the effort of experimental and clinical studies to understand the role of neuroinflammation in the underlying problem of the pathophysiology of migraine. In this review, they have mentioned about different outcomes from different experiments. The main component involved in the immune and inflammatory responses, cytokines, was got attention and examined thoroughly in this review as well as by the researchers. All the components TNF-α, IL-1ß, and IL-6 levels were found elevated in the various mice models in the pathophysiology of migraine study. On the contrary, it was observed that human blood has no change in TNF-α levels and inconsistent levels in IL-1ß and IL-6 during migraine attacks. Then, they also discussed about the saliva-based tests and their possibilities in the pathophysiology studies of migraines. Overall, the manuscript is well written. In my opinion, a minor improvement is required at the end of the manuscript in the conclusion part. Authors may state the new therapeutic avenues from the cytokine’s aspect more clearly. A clear perspective is also lacking.

Author Response

Manuscript ID: ijms-2357239

Type of manuscript: Review

Title: Experimental and clinical investigation of neuroinflammation in

migraine: A Narrative Review

To the Reviewer 2

We have revised the manuscript in accordance with the comments, and we are pleased to resubmit it. We appreciate the time and effort that the reviewer has dedicated to providing insightful feedback. All revisions and additions are indicated as yellow-highlighted text in the manuscript. We sincerely hope that with these revisions, our manuscript will be suitable for publication in the International Journal of Molecular Sciences.

Referee 2

In this manuscript titled “Experimental and clinical investigation of neuroinflammation in migraine: A Narrative Review,” the authors portrayed the effort of experimental and clinical studies to understand the role of neuroinflammation in the underlying problem of the pathophysiology of migraine. In this review, they have mentioned about different outcomes from different experiments. The main component involved in the immune and inflammatory responses, cytokines, was got attention and examined thoroughly in this review as well as by the researchers. All the components TNF-α, IL-1ß, and IL-6 levels were found elevated in the various mice models in the pathophysiology of migraine study. On the contrary, it was observed that human blood has no change in TNF-α levels and inconsistent levels in IL-1ß and IL-6 during migraine attacks. Then, they also discussed about the saliva-based tests and their possibilities in the pathophysiology studies of migraines. Overall, the manuscript is well written. In my opinion, a minor improvement is required at the end of the manuscript in the conclusion part. Authors may state the new therapeutic avenues from the cytokine’s aspect more clearly. A clear perspective is also lacking.

Response:

Thank you for your careful review and valuable comment. The following changes have been made and added to lines 445-457.

“Although the treatment approaches for migraine are not yet well established, IL1 receptor antagonists suppress migraine attacks in patients with the cryopyrin-associated periodic syndrome, where mutations in the inflammasome component NLRP3 result in over secretion of IL-1β [120,121]. Furthermore, the usefulness of anakinra has already been reported in epilepsy [122-124], in which IL-1β is deeply involved in the pathogenesis [125], and which has been pointed out to have pathophysiological similarities with migraine [126,127]. Not only IL-1β, but also TNF-α and IL-6, which are associated with migraine, as pointed out in this review, have already been established for clinical application in rheumatic diseases. The efficacy of each of these anti-cytokine therapies is also expected. As shown in Figure 1, mouse models suggest that P2X7R antagonists can inhibit these cascades [51] and prevent inflammasome complex formation [34], which will facilitate novel migraine treatment strategies. A cytokine perspective on migraine etiology may further aid the treatment of patients with migraine.”

Reviewer 3 Report

This review deals with an interesting theme, that has a significative clinical importance.

The parts regarding the research on animal models and the one about inflammation markers ‘dosing in biological fluids are accurate and exhaustive.

Anyway, considering that is a review about neuroinflammation in migraine, it is strictly necessary to review also the genetical aspects of the relationship between migraine and cytokines. The small mention of the mutations in the inflammasome component NLRP3’s gene is not enough. Indeed, it is known that some polymorphisms of cytokines’ genes can influence frequency, severity of migraine, and clinical response to migraine therapy. I suggest deepening this aspect.

The author can consider for example the following papers:

- Lee et al. Association between a polymorphism in the lymphotoxin-a promoter region and migraine. Headache 2007.

- Schürks M et al. Tumor necrosis factor gene polymorphisms and migraine: a systematic review and meta-analysis. Cephalalgia 2011.

- Gu L et al. The TNF-α-308G/A polymorphism is associated with migraine risk: a meta-analysis. Exp Ther Med 2012.

These 3 studies deal with the possible association between TNF-alpha polymorphisms and migraine, which is very discussed. The first show an association between homozygosity for the LTA-294C allele and migraine. The second is a metanalysis of previous study and refuses an association between TNFα and TNFβ gene variants and migraine. The other metanalysis (Gu et al) reports an association between TNF-α -308G/A polymorphisms and migraine risk. So, there is not a clear agreement regarding this theme, but the author should present the current situation, even with its conflicting results (as it was done for fluid cytokines’ dosing).

Other studies that can be cited are:

- Rainero I et al. A polymorphism in the interleukin-1 alpha gene influences the clinical features of migraine. Headache 2002. In this study, migraine patients carrying the T/T genotype of the IL1α gene showed a significantly lower age at disease onset; the same genotype was significantly more frequent in patients with migraine with aura than in patients with migraine without aura.

- Gormley P et al.; International Headache Genetics Consortium. Meta-analysis of 375,000 individuals identifies 38 susceptibility loci for migraine. Nat Genet 2016. This meta-analysis of 375 000 individuals, identified 38 susceptibility loci for migraine, and higlighted that several loci are linked with inflammation mechanisms, for example TGFB receptor 2.

- Rubino E et al. Polymorphisms of the Proinflammatory Cytokine Genes Modulate the Response to NSAIDs but not to Triptans in Migraine Attacks. IJMS 2023. This recent study shows that a polymorphism of TNF alpha is associated with worse response to NSAIDS.

Then I would like to suggest the following corrections regarding misprints/unclear sentences in the text:

-Line 223: “The number of studies regarding anti-inflammatory cytokines, mainly IL-4 and IL-10, is limited compared to the number of the ones regarding inflammatory cytokines”.

- Line 342: “In mouse models, CSD induces temporal alterations in inflammatory cytokines such 342 as TNF-α, IL-1ß, and IL-6, and TNF-α in particular, has been implicated in CSD” -> I think there is a repetition, so I would write “In mouse models, CSD induces temporal alterations in inflammatory cytokines such 342 as TNF-α, IL-1ß, and IL-6, and TNF-α in particular”

- Line 229: “Factors that prevent a consistent view of cytokines during migraine attacks include the number of samples, measurement methods, and the broad clinical spectrum” -> Did you mean “factors that modify cytokines’ levels include...”?

- Line 262: “As for IL-1ß levels, most interictal investigations of IL-1ß show no change compared to controls [69, 72, 82-84, 87], while some reports have shown increased levels” -> I would state simply: “most interictal investigations of IL-1ß show no change compared to controls [69, 72, 82-84, 87], while some reports have shown increased levels”. It is clearer.

- Line 280: “Most studies have reported decreased levels of IL-10 during attacks in migraine patients compared to controls, mainly” -> Did you mean outside attacks? Remove “, mainly”.

In my opinion the paper should be accepted, but only after a major revision comprising an exhaustive addendum regarding genetics.

Kind regards

Minor editing of English language required

Author Response

Manuscript ID: ijms-2357239

Type of manuscript: Review

Title: Experimental and clinical investigation of neuroinflammation in

migraine: A Narrative Review

To the Reviewer 3

Thank you for your careful review and valuable comments. We appreciate the time and effort that the reviewer has dedicated to providing insightful feedback. We sincerely hope that with these revisions, our manuscript will be suitable for publication in the International Journal of Molecular Sciences. All revisions are indicated as yellow-highlighted text in the manuscript.

Comments

Referee3

Main comments

This review deals with an interesting theme, that has a significative clinical importance.

The parts regarding the research on animal models and the one about inflammation markers ‘dosing in biological fluids are accurate and exhaustive.

Anyway, considering that is a review about neuroinflammation in migraine, it is strictly necessary to review also the genetical aspects of the relationship between migraine and cytokines. The small mention of the mutations in the inflammasome component NLRP3’s gene is not enough. Indeed, it is known that some polymorphisms of cytokines’ genes can influence frequency, severity of migraine, and clinical response to migraine therapy. I suggest deepening this aspect.

The author can consider for example the following papers:

- Lee et al. Association between a polymorphism in the lymphotoxin-a promoter region and migraine. Headache 2007.

- Schürks M et al. Tumor necrosis factor gene polymorphisms and migraine: a systematic review and meta-analysis. Cephalalgia 2011.

- Gu L et al. The TNF-α-308G/A polymorphism is associated with migraine risk: a meta-analysis. Exp Ther Med 2012.

These 3 studies deal with the possible association between TNF-alpha polymorphisms and migraine, which is very discussed. The first show an association between homozygosity for the LTA-294C allele and migraine. The second is a metanalysis of previous study and refuses an association between TNFα and TNFβ gene variants and migraine. The other metanalysis (Gu et al) reports an association between TNF-α -308G/A polymorphisms and migraine risk. So, there is not a clear agreement regarding this theme, but the author should present the current situation, even with its conflicting results (as it was done for fluid cytokines’ dosing).

Other studies that can be cited are:

- Rainero I et al. A polymorphism in the interleukin-1 alpha gene influences the clinical features of migraine. Headache 2002. In this study, migraine patients carrying the T/T genotype of the IL1α gene showed a significantly lower age at disease onset; the same genotype was significantly more frequent in patients with migraine with aura than in patients with migraine without aura.

- Gormley P et al.; International Headache Genetics Consortium. Meta-analysis of 375,000 individuals identifies 38 susceptibility loci for migraine. Nat Genet 2016. This meta-analysis of 375 000 individuals, identified 38 susceptibility loci for migraine, and higlighted that several loci are linked with inflammation mechanisms, for example TGFB receptor 2.

- Rubino E et al. Polymorphisms of the Proinflammatory Cytokine Genes Modulate the Response to NSAIDs but not to Triptans in Migraine Attacks. IJMS 2023. This recent study shows that a polymorphism of TNF alpha is associated with worse response to NSAIDS. {Rubino, 2022 #2914}

Response: 

Thank you for your careful review and valuable comments. We have made the following corrections:

  1. Cytokines and Genetics (Lines 345-406)

“As mentioned above, animal and human studies suggest that TNF-α contributes to the pathogenesis of migraine and that its secretory capacity might be regulated by a gene variant in the promoter region of the TNF gene. Guanine, located upstream of 308 base pairs in the TNF gene (NC_000006.12:g.31575254G>A), which is replaced by adenine, is thought to increase circulating levels of TNF-α[103]. Variants registered in dbSNP as rs1800629 have been noted as being associated with migraine, with some reports showing that migraine without aura is significantly more prevalent than in healthy populations [104-106]. There are also reports that found no significant differences [107,108]. A me-ta-analysis conducted by Schürks et al. in response to this discrepancy showed no overall association between rs1800629 and migraine. Subgroup analysis showed that the risk of migraine without aura due to this variant varied by ethnicity. The risk of migraine with aura was higher among Asians, and that both effects were stronger in women [109]. Other meta-analyses showed that rs1800629 was associated with migraine overall in Asians. Subgroup analyses showed significantly higher odds ratios for variant retention in mi-graine with aura and no significant differences in migraine without aura [104]. It should be noted that the discrepancy in the results of these studies may indicate differences in the genetic background of migraine by race. An analysis of sites in linkage disequilibrium with variants including rs1800629, aimed at resolving previously reported discrepancies, showed that the LTA gene haplotype was associated with migraine risk in Koreans [110]. In addition, pharmacogenetics showed that the efficacy of NSAIDs was significantly low-er in migraine attacks in rs1800629 carriers [111], suggesting that rs1800629 may be deeply involved in the pathophysiology of migraine attacks as well as in the risk of mi-graine. The reproducibility of the results is expected to be reported.

In a study of the IL-1α gene, holders of genotype T/T in the promoter region (889 base pairs upstream) have a significantly earlier onset of migraine than the wild types C/C and C/T, which are observed more frequently in migraine with aura [112]. There is a report of significant IL-1β gene variants [106], but no significant differences between migraine and controls for the promoter variants of the IL6 gene [113] and IL10 gene [114] have been de-termined. While the results from these genetic studies are very interesting, the details of the impact of the above variants on neuroinflammation in migraine remain unclear, as the genetic variants were not designed to allow examination of their impact on neuroinflam-mation.

With the widespread use of next-generation sequencers, many Genome-wide associ-ation studies (GWAS) have been conducted for complex genetic diseases. In migraine, GWAS have been performed to clarify its genetic background: in 2016, Gormley et al. ex-plored disease susceptibility genes in 59,674 migraine patients and 316,078 controls in a meta-analysis of 22 GWAS studies [115]. They identified 38 genomic loci and showed that they were associated with genes expressed in blood vessels and smooth muscle. Among them, the TGFBR2 gene is interesting because it is a receptor for TGF-β and has implica-tions in the neuroinflammatory pathway. The fact that this GWAS meta-analysis is lim-ited to the genes most proximal to the relevant single nucleotide polymorphism (SNPs) opens the possibility that the distal genes are the true causative genes.

In 2022, Hautakangas et al. published a GWAS meta-analysis of 102,084 migraine cases and 771,257 controls that added data from four more GWAS studies to the Gormley et al. meta-analysis [116]. A total of 123 genomic loci were identified, 86 of which were novel locations. These included the CALCA and CALCB genes, which encode CGRPs tar-geted by migraine drugs, and the HTR1F gene, which encodes the 5-HT1F receptor. Inter-estingly, inflammation-related genes included the TGFBR2 gene as well as the MAP-KAPK2 and PTK2 genes, which encode MK2, a downstream substrate of the p38MAPK signaling pathway that regulates apoptosis and inflammatory responses. It has been re-ported to promote the production of TNF-α and IL-6 [117]. Pyk2 encoded by the PTK2 gene is thought to activate MAPKs and NF-kB in response to both TNF-α and IL-1β [118]. Thus, it is suggested that Pyk2 may modulate neuroinflammation in migraine in association with TNF-α.

In these GWAS meta-analyses, all subjects were of European ethnicity. Therefore, caution should be exercised when applying the results of the meta-analyses to other racial groups. For example, the results may not be consistent with the observation that TNF gene variants are more common in Asian patients with migraine.

 In human and mouse studies, there is no doubt that cytokine-mediated inflamma-tory phenomena occur during the pathogenesis of migraines.  ”  

The following was added to the 'Conclusion’ (Lines 437-4444)

“Genetic analysis of cytokines suggests that TNF-α and other cytokines are involved not only in the risk of migraine [104-106,119] but also in the pathophysiology during attacks [111]. Furthermore, comprehensive genetic analysis using GWAS has identified not only genes encoding CGRP and 5-HT1F receptors but also genes involved in the production of TNF-α and IL-6 and regulating apoptosis and inflammatory responses as candidate genes for migraine [115,116]. Although definitive results have yet to be obtained and have yet to prove a link to neuroinflammation via cytokines, progress in this field of genetic analysis is impressive and future perspectives are promising.”

Minor comments

Then I would like to suggest the following corrections regarding misprints/unclear sentences in the text:

-Line 223: “The number of studies regarding anti-inflammatory cytokines, mainly IL-4 and IL-10, is limited compared to the number of the ones regarding inflammatory cytokines”.

Response: We have made the following correction (Lines 225 and 226).

“The number of studies on anti-inflammatory cytokines, mainly IL-4 and IL-10, are fewer and more limited than the number of studies on inflammatory cytokines.”

- Line 342: “In mouse models, CSD induces temporal alterations in inflammatory cytokines such as TNF-α, IL-1ß, and IL-6, and TNF-α in particular, has been implicated in CSD” -> I think there is a repetition.

Response: I have made the following correction as you indicated (Lines 406 and 407).

“In mouse models, CSD induces temporal alterations in inflammatory cytokines such as TNF-α, in particular, and in IL-1ß and IL-6.”

- Line 229: “Factors that prevent a consistent view of cytokines during migraine attacks include the number of samples, measurement methods, and the broad clinical spectrum” -> Did you mean “factors that modify cytokines’ levels include...”?

Response: I have made the following revision to correct this mistake (Lines 231 and 232).

“Factors that modify cytokines’ levels include the number of samples, measurement methods, and the broad clinical spectrum.”

- Line 262: “As for IL-1ß levels, most interictal investigations of IL-1ß show no change compared to controls [69, 72, 82-84, 87], while some reports have shown increased levels” -> I would state simply: “most interictal investigations of IL-1ß show no change compared to controls [69, 72, 82-84, 87], while some reports have shown increased levels”. It is clearer.

Response: I have made the following correction (Lines 264 and 265).

“Most interictal investigations of IL-1ß show no change compared to controls [69, 72, 82-84, 87], while some reports have shown increased levels [67, 70, 93].”

- Line 280: “Most studies have reported decreased levels of IL-10 during attacks in migraine patients compared to controls, mainly” -> Did you mean outside attacks? Remove “, mainly”.

Response: Thank you for pointing this out. I meant ‘outside attacks,’ not during attacks. I omitted “mainly.” I have made the following correction (Lines282-283).

Most studies have reported decreased levels of IL-10 in the interictal period outside attacks in migraine patients compared to controls [67, 79, 83, 89, 93]

Thank you again for your valuable feedback.

Round 2

Reviewer 3 Report

Thank you very much for the implementation of your paper, I think that now is very complete 

Only minor english correction are needed